# Evaluation of the Parasitism Capacity of a Thelytoky Egg Parasitoid on a Serious Rice Pest, *Nilaparvata lugens* (Stål)

**DOI:** 10.3390/ani13010012

**Published:** 2022-12-20

**Authors:** Longqing Shi, Dawei Liu, Liangmiao Qiu, Zhaowei Jiang, Zhixiong Zhan

**Affiliations:** 1Rice Research Institute, Fujian Academy of Agricultural Sciences, Cangshan, Fuzhou 350018, China; 2State Key Laboratory of Ecological Pest Control for Fujian and Taiwan Crops, Institute of Applied Ecology, Fujian Agriculture and Forestry University, Fuzhou 350002, China; 3Institute of Plant Protection, Fujian Academy of Agricultural Sciences, Fuzhou 350013, China

**Keywords:** *Pseudoligosita yasumatsui*, *Anagrus nilaparvatae*, biological control, parthenogernesis, rice planthopper

## Abstract

**Simple Summary:**

*Nilaparvata lugens* (Stål) (BPH) is a major rice pest that severely reduces global rice production. Egg parasitoids can eliminate BPH eggs and effectively reduce the damage caused by BPH. In this study, we assessed the parasitism capacity of a strain of *Pseudoligosita yasumatsui* reproduced via thelytoky, which was compared to *Anagrus nilaparvatae*, one of the dominant egg parasitoid of rice planthoppers. The results showed that both egg parasitoids preferred parasitizing the fertilized BPH eggs and that parasitization mostly occurred during the daytime, notably during 07:00–15:00. The parasitism capacity of this strain of *P. yasumatsui* was slightly higher than that of *A. nilaparvatae*. Both the egg parasitoids preferred parasitizing 1–3-day-old BPH eggs. However, the parasitism of this strain of *P. yasumatsui* on older BPH eggs was better than that of *A. nilaparvatae*. The findings support the use of *P. yasumatsui* to control the rice pest *N. lugens*.

**Abstract:**

*Pseudoligosita yasumatsui* and *Anagrus nilaparvatae* are both egg parasitoids of the brown planthopper, *Nilaparvata lugens* (Stål) (BPH). In this study, we obtained a stable strain of *P. yasumatsui* reproduced via thelytoky through indoor rearing and screening. We assessed the parasitism capacity of this strain on eggs of *N. lugens* by comparing the parasitism preference and circadian rhythm of this strain to that of *A. nilaparvatae*, which is proved as the dominant egg parasitoid species of BPH in rice fields. The findings indicated that both egg parasitoids could parasitize fertilized and unfertilized BPH eggs, however, with a significant preference for fertilized eggs. The daily parasitization volume of *P. yasumatsui* was slightly higher than that of *A. nilaparvatae*. Both egg parasitoids preferred parasitizing 1–3-day-old BPH eggs, but the parasitism amount of 5–6-day-old BPH eggs by *P. yasumatsui* is higher than that by *A. nilaparvatae*. The parasitism events of both species of egg parasitoid wasps occurred primarily from 7:00–15:00 and the parasitism amount at night accounted for less than 15% of the total amount. The results indicate that this strain of *P. yasumatsui* reproduced via thelytoky could be valuable for rice planthopper control.

## 1. Introduction

As one of the world’s major food crops, rice is the staple food of nearly half of the global population [1]. *Nilaparvata lugens* (Stål) (Hemiptera: Delphacidae) (BPH) is a major pest in rice fields, especially in Asia, and its rapid reproduction rate can quickly lead to outbreaks, severely reducing rice yield and quality [2,3]. Currently, the methods for controlling BPH include using chemical insecticides and the natural enemies, i.e., egg parasitoids. The latter has received extensive attention owing to its green and sustainable characteristics [4]. Egg parasitoids grow and develop by laying eggs in the host eggs and feeding on the nutrients, resulting in the death of the host eggs because they cannot develop properly. The Mymaridae and Trichogrammatidae families have been reported to be the primary egg parasitoids of rice planthoppers [5]. *Anagrus nilaparvatae* (Pang and Wang) (Hymenoptera: Mymanidae) and *Pseudoligosita yasumatsui* (Hymenoptera: Trichogrammatidae) are the primary BPH egg parasitoids [6]. Compared with other egg parasitoids, *A. nilaparvatae* is the dominant species of BPH egg parasitoid with high fecundity, large population size, wide ecological niche, and strong ecological adaptability [7,8]. The parasitism rate of *A. nilaparvatae* on BPH eggs in rice fields is maintained at approximately 70% [9] and is a primary biological factor for inhibiting the growth of BPH populations [10]. While *P. yasumatsui* is prevalent in the rice ecology, its population is smaller than that of *A. nilaparvatae* [11,12]. Both *A. nilaparvatae* and *P. yasumatsui* can reproduce via gamogenesis and parthenogenesis [6,13,14]. Gamogenesis requires mating between males and females, fertilization of egg cells, and egg laying by females to produce fertile offspring. In contrast, parthenogenesis requires no mating between male and female and egg cells in the ovaries of the females develop directly into new individuals without fertilization. *A. nilaparvatae* can reproduce via arrhenotoky and thelytoky has not been reported. Trichogrammatidae also can reproduce via arrhenotoky, in which most strains still produce male offspring [15]. However, some other strains affected by the symbiotic bacteria or other factors can reproduce via thelytoky, producing female offspring via parthenogenesis [13,16,17]. Compared with gamogenesis and parthenogenesis/arrhenotoky, thelytoky boasts better application prospects owing to its advantages of high multiplication capacity, easy colonization, and low production cost in the field of biological pest control [15,18]. Trichogrammatidae is one of the most successfully applied egg parasitoids in biological control [19]. Trichogrammatidae can parasitize eggs of over 1270 insect species that belong to 90 families and 11 orders, including Lepidoptera, Hymenoptera, and Diptera. Trichogrammatidae-based products have been developed to control pests on corn, rice, vegetables, and forest and fruit trees [20,21,22]. Trichogrammatidae are used to control moth pests that affect rice crops [23]; however, Trichogrammatidae that can be applied to parasitize rice planthopper eggs are unavailable. Through years of indoor-rearing and screening, we obtained a strain of *P. yasumatsui* reproduced via thelytoky and all offspring reproduced via parthenogenesis were female with stable traits. In this study, the parasitism capacity and application prospects of this strain of *P. yasumatsui* were assessed using the dominant species of BPH egg parasitoids, with *A. nilaparvatae*, as a control.

## 2. Materials and Methods

### 2.1. Experimental Materials

All the test insects were obtained from the Rice Research Institute of Fujian Academy of Agricultural Sciences (Fuzhou, Fujian, China). Both *A. nilaparvatae* and the *P. yasumatsui* strain reproduced via thelytoky were maintained using BPH eggs as hosts for over 10 generations. The rice seedlings used in this study were all (Taichung Native 1, TN1) varieties. The temperature of the artificial climate chamber was set at 28 ± 1 °C, relative humidity at 70 ± 5%, with a photoperiod of L:D = 14:10. Additionally, all the below experiments were performed under the conditions of the same temperature, relative humidity, and photoperiod.

### 2.2. Comparison between Parthenogenesis and Sexual Reproduction of A. nilaparvatae

A two-leaf rice seedling was placed in a transparent PVC tube (length, 10 cm; diameter, 2 cm), and three adult impregnated female BPH were placed in each tube to lay eggs on the rice seedling. After 24 h, one unmated adult female *A. nilaparvatae* within 12 h of emergence was introduced. The survival of *A. nilaparvatae* was observed daily. After the death of *A. nilaparvatae* in the tube, the rice seedlings were removed and the BPH eggs were obtained and incubated in a Petri dish lined with moist filter paper. The number of parasitized BPH eggs and the number of males and females of *A. nilaparvatae* offspring after emergence were recorded. Ten replicates were established for this experiment.

In a separate experiment, a pair of male and female *A. nilaparvatae* was selected within 12 h of emergence. Ten replicates were set up to record the number of parasitized eggs of *N. lugens* and the number of males and females of the offspring *A. nilaparvatae* after emergence.

### 2.3. Parasitism Differences between Two Types of Egg Parasitoids on Fertilized/Unfertilized BPH Eggs

Ten PVC tubes (10 cm in length and 2 cm in diameter) were prepared and each PVC tube contained a two-leaf rice seedling and a female adult BPH. For five of the ten tubes, the BPH female was unmated and lived 5 days after emergence while the other five contained a mated female adult and lived 5 days after emergence. After removing BPH from the PVC pipes after 24 h, we obtained the rice seedlings with plenty of unfertilized or fertilized BPH eggs. The 10 rice seedlings were arranged at intervals, as presented in Figure 1. In this experiment, two treatment groups were set up: 10 pairs of *A. nilaparvatae* (male-to-female ratio of 1:1) were introduced into Treatment group 1 and 10 female *P. yasumatsui* were introduced into Treatment group 2. After 24 h, all the test insects were removed from both treatments. The rice seedlings were removed after 3 days. The BPH eggs inside the rice seedlings were found and incubated in Petri dishes lined with moist filter papers. The number of total parasitized BPH eggs was counted. This experiment was set up in triplicate.

### 2.4. Parasitism Differences between Two Species of Egg Parasitoids on BPH Eggs at Different Egg Ages

To determine the developmental status of fertilized BPH eggs at different egg ages, a two-leaf rice seedling was prepared, and an impregnated adult BPH female was introduced to lay eggs on the seedling. The BPH were removed 24 h later and the eggs on the seedling were found and incubated in a Petri dish lined with moist filter papers. The temperature was set to 28 ± 1 °C and the photoperiod was set at L:D = 14:10. On the day of harvesting the BPH eggs, they were placed under a stereomicroscope (OLYMPUS SZ61 microscope + DigiRetina 16 camera) and photographed every 24 h to record the development of the eggs until they hatched.

Seven square leaf boxes (5 cm × 5 cm × 8 cm) were prepared and 10 TN1 rice seedlings were planted in each box until they reached the two-leaf stage. Ten impregnated adult BPH females at 5 days after emergence were introduced into the first box to lay eggs. After 24 h, the females were removed. The BPH eggs at 1–7 days of egg age were obtained from these steps and the boxes of rice seedlings with BPH eggs were arranged in a circle, as presented in Figure 2. Depending on the treatment, 10 pairs of adult *A. nilaparvatae* within 12 h of emergence or 10 adult female *P. yasumatsui* within 12 h of emergence were introduced into the center of the circle and the egg parasitoids were removed after 24 h. The BPH eggs in the rice seedlings were found after 3 days and the numbers of eggs laid and parasitized were recorded. Two treatment groups were set up according to the species of egg parasitoids and three replications were conducted for each group.

### 2.5. Differences in the Parasitization Period of the Two Species of Egg Parasitoids

At 07:00, 35 impregnated adult BPH females were placed on 35 two-leaf rice seedlings to lay eggs. After 24 h, the females were removed and the rice seedlings were divided into 7 groups of 5 seedlings. The groups of seedlings were removed at various times for egg parasitoids to parasitize. The first group of rice seedlings was placed in an insect cage (80 cm × 40 cm) at 7:00 on the same day and egg parasitoids of corresponding numbers were introduced. After the first group of seedlings was placed, the rice seedlings were changed with a new group every 2 h. The rice seedlings parasitized by egg parasitoids were transferred to another insect cage for further cultivation. At 19:00, the seventh group of rice seedlings was placed in and not changed until 7:00 on the following day, when they were removed to another insect cage. Three days later, all the BPH eggs were removed from each group of rice seedlings via dissection and the number of eggs and the parasitism amount were counted. Two treatments were set up according to the species of egg parasitoids: 10 pairs of *A. nilaparvatae* within 12 h of emergence or 10 adult *P. yasumatsui* females within 12 h of emergence were placed in each treatment. Three replications of each treatment were conducted as described above.

### 2.6. Data Analysis

All the data were initially compiled using Microsoft Excel 2016. Firstly, all the data were analyzed using Jamovi Statistical Software to determine the normality, independence, and homogeneity of variance. The data of parthenogenesis and sexual reproduction of *A. nilaparvatae* were analyzed using independent *t*-tests. The data on the parasitism selection of fertilized and unfertilized BPH eggs by two species of egg parasitoids were analyzed using *t*-tests. The data on the selection by the two species of egg parasitoids on fertilized and unfertilized eggs and at day/night were analyzed using *t*-tests. The data on the parasitism preference of the two egg parasitoids on 1–7 days BPH eggs and in daytime (07:00–19:00) were analyzed using one-way ANOVA and multiple comparisons (Tukey’s honest significant difference, Tukey’s HSD). All the above were considered significant differences at *p* < 0.05.

## 3. Results

### 3.1. Comparison of the Differences between the Two Reproductive Modes of A. nilaparvatae

In the absence of nutritional supplementation, the average survival of *A. nilaparvatae* in this experiment was about 2 days. When the male and female *A. nilaparvatae* were paired, the average parasitism amount of BPH eggs was 9.3. Moreover, the offspring contained males and females, with females outnumbering males (male ratio 0.26). When a single *A. nilaparvatae* female was placed at parthenogenesis, the need to spend time on courtship and mating was eliminated. The average parasitism amount of adult *A. nilaparvatae* females was 19.7, significantly higher than that of the male–female pairing group, but all the offspring were males (Table 1).

### 3.2. Comparison of the Parasitism Amount of Fertilized and Unfertilized Eggs by Two Species of Egg Parasitoids

In the presence of excess BPH eggs, the average parasitism amount of 10 *A. nilaparvatae* adult females (1:1 male:female pair) on fertilized BPH eggs for 24 h was 66.6 (Table 2), which was significantly higher than the average parasitism amount of 25.4 on unfertilized BPH eggs. Moreover, the parasitism amount of 10 *P. yasumatsui* females on fertilized BPH eggs was 68.4, while the parasitism amount of unfertilized BPH eggs was 40.2, which was significantly higher than that of unfertilized BPH eggs. The parasitism amount of the fertilized BPH eggs was significantly higher than that of the unfertilized eggs. However, all the parasitoid eggs laid in unfertilized BPH eggs could develop normally into adults. The findings indicate that both species of egg parasitoid wasps preferred parasitizing fertilized BPH eggs.

### 3.3. Parasitism Preference of the Two Species of Egg Parasitoids for BPH Eggs of Different Egg Ages

The BPH eggs were developed in the conditions of 28 ± 1 °C, relative humidity at 70 ± 5%, with a photoperiod of L:D = 14:10. After the impregnated BPH females laid eggs, the eggs begin to develop at a suitable temperature and it generally required around 9 days for the first instar nymphs to hatch. On the day the BPH eggs were laid, the eggs entered the placental period (Figure 3a): the eggs were grayish white, translucent, with evenly distributed inclusions, and a uniformly elongated egg shape, with a waxy white spot at the posterior end of the egg. From day 2 onwards, the eggs entered the embryonic band formation period, with scale-like inclusions in the eggshell (Figure 3b) and band formation, while the chambers within the eggs began to form. On day 3 of egg development, the eggs entered the embryo formation stage, with evident color changes and waxy white or yellowish patches at the anterior end of the eggs (Figure 3c), and the embryo gradually took shape. On day 4, the eggs entered the embryo movement and embryo segmentation stages, at which time the embryo segments took shape and small red eyespots appeared at the head end of the eggs (Figure 3d). From days 5–7, the eggs entered the appendage stage: the eyespots at the anterior end of the eggs were blood-clotted (Figure 3e–g). At that time, the thoracic appendages were revealed and further developed into thoracic legs (Figure 3h). On day 9, the nymph bit through the eggshell and hatched (Figure 3i).

The amount of BPH eggs parasitized by *A. nilaparvatae* or *P. yasumatsui* both decreased with the increasing egg age (Figure 4) and the highest parasitism amount happened in the 1–4-days-old BPH eggs. The findings of multiple comparisons indicated that the *A. nilaparvatae* parasitism amount of 1- or 2-days-old BPH eggs was significantly higher than that of 3–6-days-old BPH eggs. As for *P. yasumatsui*, the parasitism amount of the 1–3-days-old BPH eggs was significantly higher than that of 4–6-days-old BPH eggs. Overall, both species of egg parasitoids were able to parasitize 1–6-days-old BPH eggs and preferred parasitizing 1–3-days-old BPH eggs. However, the parasitism amount of 5–6-days-old BPH eggs by *P. yasumatsui* remained considerable compared to the extremely low parasitism amount of the 5–6-days-old BPH eggs by *P. yasumatsui*.

### 3.4. Parasitism Rhythm of the Two Species of Egg Parasitoids

As shown in Table 3, the parasitism amount of 10 *A. nilaparvatae* accounted for 87.5% of the entire day during the daytime period (7:00–19:00) and for a mere 12.5% during the nighttime period (19:00–7:00). In contrast, the parasitism amount of 10 *P. yasumatsui* accounted for 92.5% of the entire day during the daytime period (7:00–19:00) and for a mere 7.5% during the nighttime period (19:00–7:00). Further analysis of the daytime period (7:00–19:00) indicated that the parasitism activity of *A. nilaparvatae* occurred primarily between 7:00 and 15:00 and the parasitism amount was lower between 15:00 and 19:00 (Figure 5). In contrast, the parasitism activity of the *P. yasumatsui* was primarily between 7:00 and 13:00 and the parasitism amount was lower in the remaining period (13:00–19:00).

## 4. Discussion

Egg parasitoids are important natural enemies of pests and are key biocontrol factors for controlling rice planthopper populations [6,24]. Based on the above findings, we considered that *A. nilaparvatae* males play a vital role in the reproduction of their populations. Therefore, in this study, male–female paired *A. nilaparvatae* were used as a control for this strain of *P. yasumatsui* reproduced via thelytoky. In this study, both the *P. yasumatsui* and *A. nilaparvatae* were solitary parasitism egg parasitoids and one BPH egg was rarely parasitized by two or more egg parasitoids offspring that developed simultaneously. Both species of these egg parasitoids were developed in BPH eggs from egg to emergence as an adult. The *A. nilaparvatae* started to appear orange-red or yellow in the middle and late larvae phases of development, while the *P. yasumatsui* was earthy yellow. Thus, the parasitized BPH eggs could be distinguished by color (authors’ observation). In this study, this strain of *P. yasumatsui* reproduced via thelytoky preferred parasitizing fertilized BPH eggs, slightly higher than the dominant species of *A. nilaparvatae*. The average parasitism amount of BPH eggs at different egg ages decreased significantly with the increase in egg age. They exhibited a certain parasitism amount for BPH eggs at 5–6 days of age. The findings of the above studies suggested that this strain of *P. yasumatsui* reproduced via thelytoky has the potential to be a dominant species for BPH egg parasitoid wasps in rice fields.

Most egg parasitoids such as *Telenomus remus, Trichogramma japonicum, P. yasumatsui,* and *Trichogramma leucaniae* parasitize more fertilized host eggs than unfertilized eggs [25,26]. The results of this study showed that both *P. yasumatsui* and *A. nilaparvatae* preferred parasitizing fertilized BPH eggs in the presence of both fertilized and unfertilized BPH eggs. Chemical information substances are one kind of the primary cues for parasitoid wasps to search for, locate, and distinguish between hosts. During egg laying and damage to the plant, the secretions produced by pests can also induce the production of oviposition-induced plant volatiles (OIPVS) that attract egg parasitoids [27,28]. Hosts themselves can also attract egg parasitoids by their volatile odors. For example, egg parasitoids of the *genus Trissolcus sp*. are attracted through the odor of impregnated females and sexually mature male hosts [29]. Moreover, host eggs laid after mating are stained with male fertilization vesicle complexes, including semen, which can be attractive to egg parasitoid wasps [30]. We suggest that the two species of egg parasitoids in this study are likely to be identified by differences in the secretions of fertilized and unfertilized BPH eggs or the resulting differences in OIPVS. However, this conjecture needs to be confirmed by further studies.

The host egg’s age also significantly affects the parasitoid selection of egg parasitoid wasps. Although *Trichogramma pretiosum* can parasitize eggs of *Diatraea grandiosella* at 96–120 h (black-head stage) of egg age, its parasitoid selection prefers the latter in the presence of concurrent eggs of smaller hosts, such red-bar stage eggs (48–72 h old) [31]. Similarly, for example, the parasitism rate of *T.chilonis* eggs of *Plutella xylostella* at 1 day egg age is significantly higher than eggs at other egg ages [32]. The results of our study also showed that BPH eggs at 1 d and 2 d of egg age were significantly favored by *P. yasumatsui* and *A. nilaparvatae* when BPH eggs of all egg ages were present simultaneously. The quality of the host, such as the nutrient content, plays a key role in the development of the parasitoids and their offspring, as the host’s nutrient richness results in larger and more fertile offspring. Conversely, the lack of host nutrition leads to smaller and less fertile offspring, which tend to be deformed and even die [33]. During long-term evolution, parasitoids will allocate more offspring to nutrient-rich hosts while allocating fewer or even no offspring to low nutritional-quality hosts [34]. As the eggs develop, the nutrients within the eggs are gradually depleted by the development of the host itself and the host embryos are not easily killed, which is detrimental for the egg parasitoid wasps to complete their development [35]. In this study, *P. yasumatsui* still had a certain parasitism selection for BPH eggs at a 5 d and 6 d egg ages, which was not favorable for the development of the *P. yasumatsui* offspring. However, from the perspective of pest control, a wider selection of host egg ages could extend the control cycle of egg parasitoids and thus is conducive to enhancing the control effect.

## 5. Conclusions

In this study, we evaluated the capacity of this strain of *P. yasumatsui* reproduced via thelytoky in terms of the parasitism amount, host egg age selection, and parasitism differences in day and night, in relation to *A. nilaparvatae*, the dominant species of BPH egg parasitoids. Moreover, the indoor assessment of the application prospects of this strain was conducted. To a certain extent, this study provides theoretical parameters for the rearing and release of *P. yasumatsui* and provides data support and innovative ideas for the use of *P. yasumatsui* for the field control of rice planthoppers.

## Figures and Tables

**Figure 1 animals-13-00012-f001:**
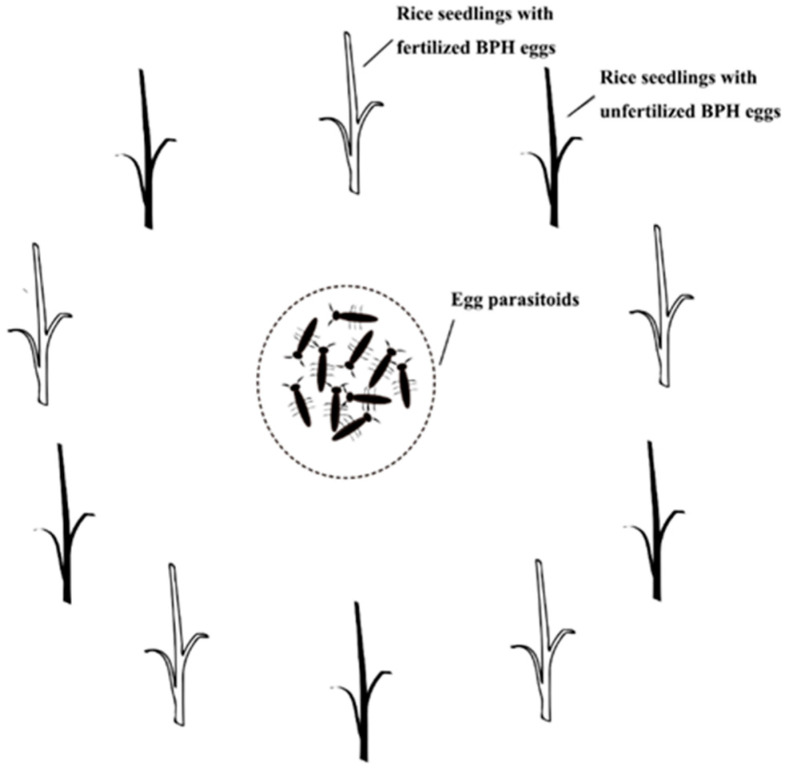
Experimental illustration of the parasitism preference of egg parasitoids between fertilized and unfertilized BPH eggs.

**Figure 2 animals-13-00012-f002:**
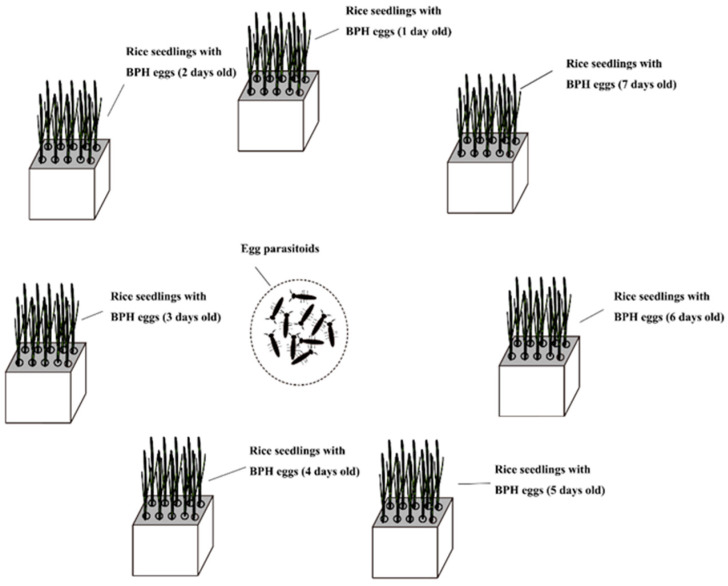
Experimental illustration of the parasitism preference of egg parasitoids among 1–7-days-old BPH eggs.

**Figure 3 animals-13-00012-f003:**
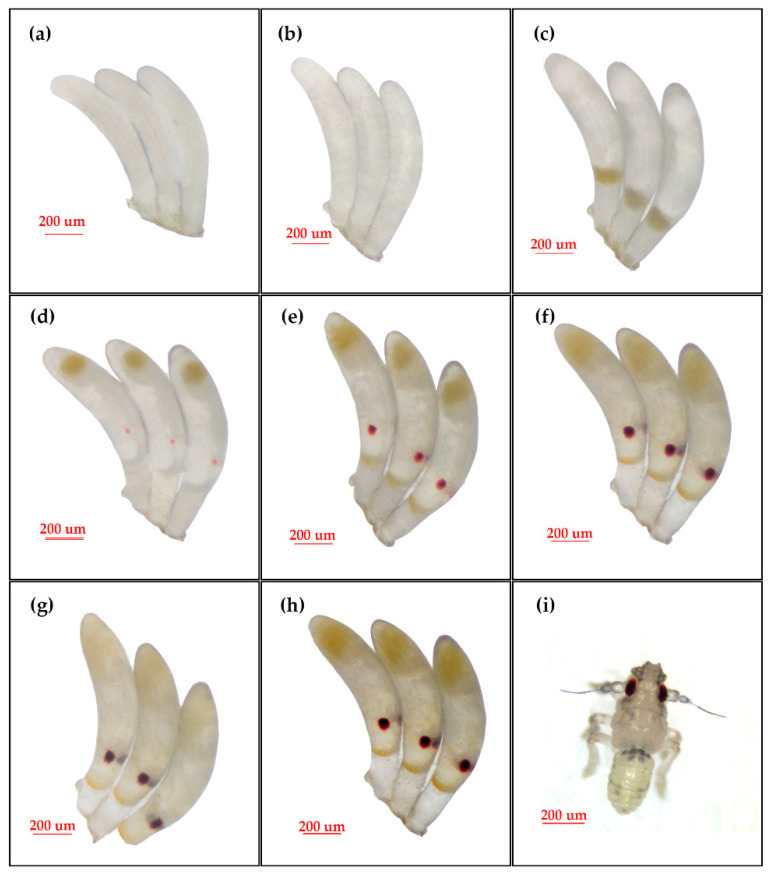
Development process of BPH eggs. (**a**–**h**), the 1–8 days ages of BPH fertilized eggs; (**i**), the new hatched BPH nymph.

**Figure 4 animals-13-00012-f004:**
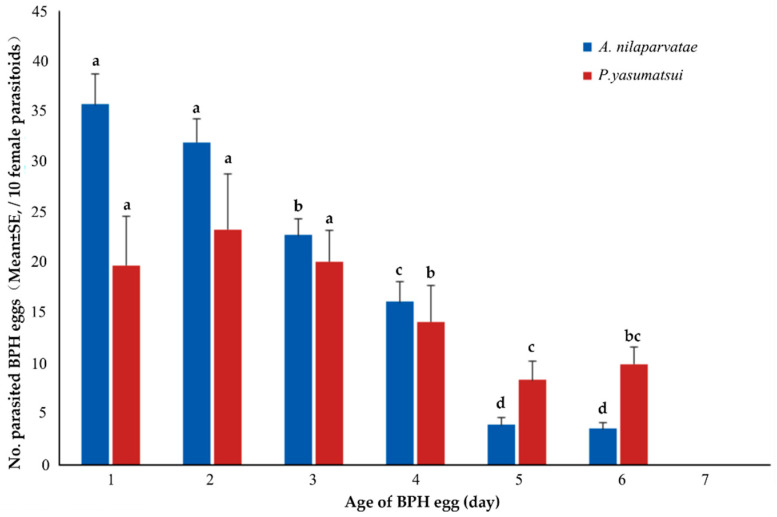
Parasitism preference of the two egg parasitoids among different ages of BPH eggs. Significant differences of the same egg parasitoid are marked with different letters (Tukey’ HSD, *p* < 0.05).

**Figure 5 animals-13-00012-f005:**
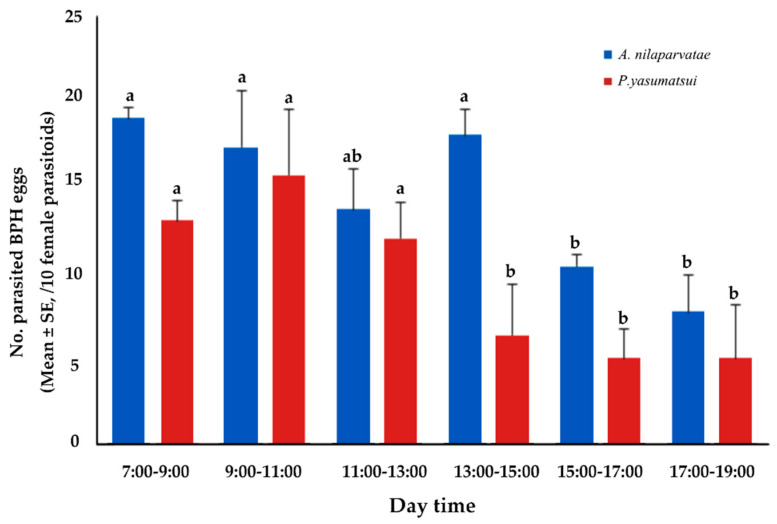
Distribution of parasitism events of the two egg parasitoids during daytime. Significant differences of the same egg parasitoid are marked with different letters (Tukey’ HSD, *p* < 0.05).

**Table 1 animals-13-00012-t001:** Comparison of parasitism ability of female *A. nilaparvatae* with/without male participation.

Treatments	No. Parasited BPH Eggs (Mean ± SE, /1 *A. nilaparvatae* Female)	Ratio of Male *A. nilaparvatae* Offsprings (Mean ± SE)
1 male + 1 female	9.3 ± 3.08 b	0.26 ± 0.030 b
1 female	19.7 ± 3.74 a	1.00 ± 0.000 a

Significant differences in the same column are marked with different letters (*t*-test, *p* < 0.05).

**Table 2 animals-13-00012-t002:** Parasitism preference of the two egg parasitoids between fertilized and unfertilized BPH eggs.

Egg Parasitoids	BPH Eggs	No. Parasited BPH Eggs (Mean ± SE)
*A. nilaparvatae*	Fertilized egg	66.6 ± 11.40 a
	Unfertilized egg	25.4 ± 7.43 b
*P. yasumatsui*	Fertilized egg	68.4 ± 6.97 a
	Unfertilized egg	40.2 ± 6.44 b

Significant differences of the same egg parasitoid species are marked with different letters (*t*-test, *p* < 0.05).

**Table 3 animals-13-00012-t003:** Parasitism activity of the two egg parasitoids during day and night.

Egg Parasitoids	Time	Ratio of Parasite BPH Egg
*A. nilaparvatae*	Daytime (7:00–19:00)	0.875 ± 0.011 a
	Nighttime (19:00–7:00)	0.125 ± 0.011 b
*P. yasumatsui*	Daytime (7:00–19:00)	0.925 ± 0.024 a
	Nighttime (19:00–7:00)	0.075 ± 0.024 b

Significant differences of the same egg parasitoid are marked with different letters (*t*-test, *p* < 0.05).

## Data Availability

The data presented in this study are available on request from the corresponding author.

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
