# Peer review of "Evaluation of the Parasitism Capacity of a Thelytoky Egg Parasitoid on a Serious Rice Pest, Nilaparvata lugens (Stål)"

_animals, 2022, doi:10.3390/ani13010012_

Round 1

Reviewer 1 Report

In this study, “Evaluation of the Parasitic Capacity of a Thelytoky Egg Parasitoid on a serious rice pest, Nilaparvata lugens (Stål)”, the authors mainly assessed the parasitic ability of the P. yasumatsui strain and illustrated the potential application of biological control against the brown planthopper.

The authors compare some important traits between P. yasumatsui and A. nilaparvatae, regarding the amount of parasitism, host egg age selection, and parasitization diurnal rhythm. In addition, the authors provide some parameters for the indoor rearing of egg parasitoids and the useful idea of related experiments.

In my comments below, I provide a more detailed overview of some major areas of potential improvement for the paper as well as several minor issues.

 1. In result 3.2, the authors report that the parasitic amount of the parasitized unfertilized eggs was higher by P. yasumatsui compared to A. nilaparvatae, which seems to suggest that P. yasumatsui has higher egg-laying capacity than A. nilaparvatae within the refined time(12 h and 24 h). So, is there any difference in the reproductive ability between them?

2. Although authors report that P. yasumatsui had a less strong preference concerning BPH egg ages than A. nilaparvatae, I believe that the implication of the more adaptable range is overreached(line 253), and the same concern (line 295).

3. In figures 4 and 5, the reproductive ability of P. yasumatsui seems to be lower than A. nilaparvatae, which is not consistent with the result 3.2. At the same time, the authors did not offer any explanation why different numbers of BPH eggs were parasitized in different analyses, which made the related parts hard to understand(acknowledge).

4. L 206, There is a confusedly-spelling mistake in the” unfertilized BPH eggs was significantly higher than that of unfertilized eggs.”

Reviewer 2 Report

Manuscript ID: animals-2078887

Title: Evaluation of the Parasitic Capacity of a Thelytoky Egg Parasitoid on a serious rice pest, Nilaparvata lugens (Stål)

General comment

Shi et al. used Anagrus nilaparvatae, the major natural enemy of the rice planthopper Nilaparvata lugens, as a control to assess the parasitism capacity of the eggs parasitoid, Pseudoligosita yasumatsui, reproduced via thelytoky. The study gives some useful information about the capacity of this strain of P. yasumatsui reproduced via thelytoky to parasitize the eggs of N. lugens, one of the most serious pests of rice in Asia. In my opinion, the introduction seems well structured, and the experiment designs were clearly explained, but some terminology used in the Ms and some parts of the results and discussion need to be revised.

Specific comments

Simple Summary

L20-L22: Why authors did not analyse it? You could include the species variable in the analysis!

Introduction

L47: add (Hemiptera: Delphacidae) to Nilaparvata lugens (Stål)

L47: maybe clarify ... that is a plant rice pest!? and one of major pests in Asia...?

L55: Same above, add order and family, Anagrus nilaparvatae (Pang et Wang) (Hymenoptera: Mymanidae) and Pseudoligosita yasumatsui ...

Materials and Methods

I am assuming that all experiment conditions were similar to those initially described in section 2.1. Experimental Materials: 28 1 °C, 75% relative humidity, and a photoperiod of L:D 14:10. Maybe this should be clarified.

L123: remove "and"

L 170-179. 2.6. Data Analysis:

I am wondering why the authors did not carry out a two-way ANOVA, including species in the model!?

In the case of the parasitism ratio (data on proportions) comparison between day and night, I am not sure that using an ANOVA is the correct way without data transformation.

Results

L189-191: Maybe this last sentence could be in discussion; I suggest removing this part from the results section.

L201-213. 3.2. Comparison of the Parasitic…

Was the number of BPH eggs available to parasitize always the same? This should be clarified
Perhaps results could be expressed in relative terms rather than absolute terms... 

L201: How many eggs were available for each egg parasitoid? Could the number of eggs available influence parasitism rates? 

L204-206: should be P. yasumatsui females?

L206-207: unfertilized -unfertilized eggs, did not get!?

L211: … thelytoky was not significantly different … This was analyzed? In data analysis, this is not explained.

L215: Significant differences of the same egg parasitoid are marked with different letters (t-test, p0.05)

Significant differences between Fertilized and unfertilized eggs…

t-test, p0.05)? or ANOVA?

L216: 3.3. Parasitic Preference of the Two Species of Egg Parasitoids for BPH Eggs of Different Egg 216 Ages

Eggs developed as described in M&M (28 ± 1 °C…), but it's worth repeating here. 

Letters in figure 3 are represented in lower case and the text in upper case.

L238-253: This section is a bit confusing; the results are clear, but the description is a bit messy, and there are parts that are inconsistent. Please revise.

L256: eggs with different ages not fertilized and unterlilized eggs.

L259: Remove statistics on the ...

L260-262: Maybe use parasitism instead of parasitic amount; please revise the terminology, not only here but throughout the manuscript.

Also, I believe these two sentences could be combined. And for the case A. nilaparvatae too.
Please simplify the section too.

L273: I am not sure if this is the correct way to use parasitic preference. There is no choice. Maybe use parasitism activity during the day and night?   
I think authors should revise some terminologies.

L274: Significant differences of the same egg parasitoid are marked with different letters (t-test, p0.05).

Significant differences between…, t-test, p0.05)? or ANOVA?

L280: 4. Discussion

The discussion should be improved. There are some parts that are well discussed, but most of the results were slightly discussed or poorly discussed. The authors should work a bit more on this part. And the discussion focused primarily on P. yasumatsui's findings, while A. nilaparvatae results were underestimated.

L286-287: This is explained here, but not in the results, and no citations are provided. 

L291-292: this seems results explanation

L292-293: These two sentences says the same!.

L296-297: This was not statistically compared.

L319: mention the names of the smaller hosts

L321-322: Maybe here you can also mention the A. nilaparvatae

Conclusions

L337-340: This, in my opinion, is incorrect; where are the advantages and disadvantages compared?
